# Characterization of T Helper 1 and 2 Cytokine Profiles in Newborns of Mothers with COVID-19

**DOI:** 10.3390/biomedicines11030910

**Published:** 2023-03-15

**Authors:** André Luís Elias Moreira, Paulo Alex Neves da Silva, Rodrigo Saar Gomes, Mônica de Oliveira Santos, Célia Regina Malveste Ito, Lucas Candido Gonçalves Barbosa, Paula Pires de Souza, Fernanda Aparecida de Oliveira Peixoto, Isabela Jubé Wastowski, Lilian Carla Carneiro, Melissa Ameloti Gomes Avelino

**Affiliations:** 1Microorganism Biotechnology Laboratory, Institute of Tropical Pathology and Public Health, Federal University of Goiás, Goiânia 74605-050, Brazil; 2Natural Immunity Laboratory, Institute of Tropical Pathology and Public Health, Federal University of Goiás, St. 235, Setor Leste Universitário, Goiânia 74605-050, Brazil; 3Neonatology of Children’s Hospital, St. 86, 160 St. Sul, Goiânia 74083-330, Brazil; 4Neonatal Intensive Care Unit (NICU), Hospital das Clínicas, Federal University of Goiás, Goiânia 74605-020, Brazil; 5Molecular Immunology Laboratory, Goiás State University, Laranjeiras Campus, St. Prof. Alfredo de Castro, 9175, Parque das Laranjeiras, Goiânia 74855-130, Brazil; 6Department of Pediatrics, Federal University of Goiás, Goiânia 74605-050, Brazil

**Keywords:** cytokine storm, newborn, mother, COVID-19, SARS, comorbidity

## Abstract

An infectious disease caused by SARS-CoV-2, COVID-19 greatly affects the pediatric population and is 3 times more prevalent in newborns than in the general population. In newborns, the overexpression of immunological molecules may also induce a so-called cytokine storm. In our study, we evaluated the expression of cytokines in newborns admitted to a neonatal ICU whose mothers had SARS-CoV-2 and symptoms of SARS. The blood of newborns of infected and healthy mothers was collected to identify their Th1 and Th2 cytokine profiles, and via flow cytometry, the cytokines TNF-α, IFN-γ, IL-2, IL-6, and IL-10 were identified. Overexpression was observed in the Th1 and Th2 cytokine profiles of newborns from infected mothers compared with the control group. Statistical analysis also revealed significant differences between the cellular and humoral responses of the infected group versus the control group. The cellular versus humoral responses of the newborns of infected mothers were also compared, which revealed the prevalence of the cellular immune response. These data demonstrate that some cytokines identified relate to more severe symptoms and even some comorbidities. IL-6, TNF-α, and IL-10 may especially be related to cytokine storms in neonates of mothers with COVID-19.

## 1. Introduction

In December 2019, the first cases of an unknown respiratory infection were reported in Wuhan, the largest city in the Chinese province of Hubei, some of which ended in death. Using molecular techniques, researchers identified and characterized the pathogen causing the disease, namely severe acute respiratory syndrome coronavirus 2 (SARS-CoV-2), the etiological agent of coronavirus disease 2019 (COVID-19) [1].

Annual records show that each year, about 4.5 million children, most of them in developing countries, die from SARS [2]. The pediatric population is indeed the part of the population most affected by SARS worldwide, with approximately 1.9 million deaths per year, 70% of them occurring in developing countries and 30% resulting from infection [3,4,5,6,7]. Within the same population, newborns are at the greatest risk of death due to respiratory infections [8,9].

Given the prevalence of SARS-CoV-2 in pediatric patients, doubts have arisen about the effects of infection in pregnant patients, particularly whether pregnancy would endanger the pregnant woman’s life or whether the infection would affect the fetus and even cause the birth of neonates with SARS-CoV-2 [10]. On that topic, viral SARS-CoV-2 particles have been detected in placentas and amniotic fluid, which suggests contamination of the fetus during its development [11,12,13,14].

Respiratory diseases caused by viruses can largely explain the high rate of mortality and morbidity in Neonatal Intensive Care Units (NICU), emergency departments, and infirmary environments [9,15]. They are also primarily responsible for SARS in newborns, are often responsible for nosocomial outbreaks [16,17,18,19], and may have symptoms similar to sepsis [20,21,22,23]. Furthermore, hospitalizations for SARS occur up to 3 times more frequently in newborns than in older children [9]. In newborns, such diseases can even progress to acute respiratory distress syndrome and respiratory failure, accompanied by acute immune dysfunction [24].

During viral infection, the immune system produces cytokines that regulate immune responses, activate cells, and induce the elimination of pathogens [25]. However, the exaggerated production of these molecules leads to a more serious inflammatory process in COVID-19, a so-called cytokine storm. Cytokine storms are exaggerated reactions of the immune system characterized by high levels of inflammatory cytokines and the exacerbated activation of cells, such as macrophages and T lymphocytes. Cytokines such as interleukin (IL)-1, IL-5, IL-6, IL-8, IL-10, interferon gamma (IFN-γ), tumor necrosis factor alpha (TNF-α), and granulocyte macrophage colony-stimulating factor have been identified as molecules that form cytokine storms [26,27,28], in a process that is important for the successful infection of other viruses [29].

Newborns do not have developed immune systems [30], and no hard data clarify the link between their immune systems and the risk of cytokine storms during SARS-CoV-2 infection. Although it is believed that newborns are extremely dependent on their innate immune systems, high viral loads disrupt those systems and increase the production of proinflammatory cytokines, which can result in cytokine storms [31,32,33].

Against this background, the aim of our study was to evaluate the expression of cytokines in newborns admitted to an NICU who were born to mothers with SARS-CoV-2 and severe respiratory symptoms. We also sought to determine the correlation of cytokines with symptom severity, in addition to comparing cytokine levels between newborns of COVID-19 mothers and neonates of healthy mothers.

## 2. Materials and Methods

### 2.1. Ethical Aspects and Hospital Units

All procedures and protocols for sample collection and processing were submitted and approved in 16 October 2020 by the Research Ethics Committee of Hospital das Clínicas—GO in Goiânia-Goiás, Brazil (Registration No. 33540320.7.0000.5078). All parents of sick newborn patients and voluntary donors signed the informed consent form. Two hospitals participated in the study: Hospital da Criança de Goiânia and Hospital das Clínicas of Universidade Federal de Goiás.

### 2.2. Target Population

The study included 20 newborns, all no more than 28 days old, who were hospitalized in an NICU of mothers who presented with severe respiratory symptoms, tested positive for SARS-CoV-2, and were admitted to the Hospital das Clínicas da Faculdade de Medicina da Universidade Federal de Goiás in April and May 2020—that is, during the first wave of COVID-19 in Brazil. Only newborns admitted to the NICU whose mothers had COVID-19 and symptoms of SARS were included. Mothers shown to have other respiratory viruses in the RT-PCR testing were excluded. Meanwhile, the control group, recruited in 2021 and 2022, consisted of 10 healthy newborn volunteers free of viral infections and other pathogens and without any records of events that could have altered their expression of cytokines. The mothers of the control patients were healthy as well.

Five variables were used to correlate the clinical conditions and severity of the neonates: enterocolitis, early-onset neonatal sepsis, asphyxia, apnea, and respiratory distress. First, enterocolitis was diagnosed according to Bell’s classification [34]. Second, early-onset neonatal sepsis, monitored in the first 72 h of life, was detected with reference to the mother’s level of infectiousness and changes in the newborn’s clinical and laboratory tests [35]. Third, asphyxia was examined if Apgar scores at 5 min were 5 or less or if the newborn required positive pressure ventilation associated with changes in arterial blood gas analysis and intubation [36,37]. All of these criteria were applied for the neonates of mothers with COVID-19 because the healthy newborns did not have low Apgar scores or any need for positive pressure ventilation. Fourth, possible apnea was monitored amid respiratory pauses of 20 s or more, followed by bradycardia and a drop in saturation [38]. Fifth and finally, respiratory distress was diagnosed and classified according to the Silverman–Andersen retraction score [39].

### 2.3. Collection and Processing of Samples

Because our study was a prospective cohort study, peripheral blood samples were collected for immunophenotypic analysis using flow cytometry in order to assess the T helper 1 (Th1) and T helper 2 (Th2) lymphocyte profiles in the serum of both groups of newborns. One group contained blood samples of neonates of mothers shown to be positive for SARS-CoV-2 in RT-PCR testing (i.e., with a characteristic clinical picture of SARS) and who were admitted to the NICU and required mechanical ventilation. The other group contained negative control samples of newborns of healthy mothers who presented with a stable clinical condition and showed normal results on biochemical tests (i.e., hemograms), no infectious processes, and spontaneous ventilation. All collected samples were kept at 4 °C and sent to the Instituto de Patologia Tropical e Saúde Pública of Universidade Federal de Goiás, at which point they were processed and stored at −20 °C. To obtain peripheral blood samples, 5 mL venous blood samples were collected using the vacuum system with Vacutainer^®^ tubes (BD™, Franklin Lakes, NJ, USA) containing ethylenediaminetetraacetic acid K3 (0.054 mL/tube), which were later homogenized for 10 min and stored at 4 °C. For processing, 5 mL of whole blood was subjected to centrifugation at 500× *g* for 30 min at 20 °C. Plasma was collected and transferred to 1.5 mL tubes, and all processed samples were stored at −80 °C until analysis.

### 2.4. Profile Characterization of Cytokines by Flow Cytometry

The plasma samples from both groups of newborns were subjected to immunophenotypic analysis using flow cytometry, namely with the Cytometric Bead Array (CBA) kit (BD™, USA), according to the manufacturer’s instructions. All samples were evaluated by labeling with phycoerythrin, a fluorochrome. The panel of monoclonal antibodies conjugated with the fluorophore used to probe, identify, and quantify cytokines in the samples were anti-IL-2, anti-IL-6, anti-IL-10, anti-TNF-α, and anti-IFN-γ. IL-4 and IL-17A were also evaluated but were not identified during analyses.

The CBA kit was used to quantify the cytokines IL-2, INF-γ, and TNF-α (i.e., Th1 profile), as well as IL-6 and IL-10 (i.e., Th2 profile), in the plasma samples. First, five populations of beads with different fluorescence intensities, conjugated with a capture antibody specific to each cytokine, were mixed to form the CBA. Second, serum from the affected or control patients was incubated with the CBA beads for 3 h at room temperature under gentle agitation. Third, the CBA beads containing the captured cytokines were washed with wash buffer, and a phycoerythrin detection reagent was added. Fourth, the beads were read in the FL3 channel of the Guava^®^ easyCyte™ 8HT Flow Cytometer (Millipore^®^, Merck, Burlington, MA, USA), and bead populations were visualized according to their fluorescence intensities, from least to most brilliant. To obtain the standard curve, the same procedure was performed. Fifth, sample analyses, fluorescence adjustments, and data acquisition were performed on the same cytometer, and in the analyses, 30,000 events were used for each sample. Sixth and last, the fluorescence obtained was analyzed using FCS Express™ version 4.0 (De Novo Software, Glendale, CA, USA).

### 2.5. Statistical Analysis

The Shapiro–Wilk (SW) test [40] was applied to assess data normality and qualify the data as either parametric (*p* > 0.05) or non-parametric (*p* ≤ 0.05). Afterwards, the test was applied to analyze the difference in the concentration of the cytokines IL-2, IL-6, IL-10, TNF, and IFN-γ in infected patients. In light of normal results, the Kruskal–Wallis test associated with the Student–Newman–Keuls test [41] was considered correct for all possible comparisons. To gauge correlations between cytokines and clinical variables, Spearman’s nonparametric test was performed, while to compare cytokine concentrations among infected and control groups, the nonparametric Mann–Whitney *U* test for independent samples was performed [42]. Statistical analysis was performed in BioEstat^®^ 5.3 and Stata^®^ 16.0, with significance set at 5%. An association analysis was also performed in a simple logistic regression test with dichotomous and continuous data, which enabled the analysis of all variables. Statistical analysis was also conducted using Jamovi^®^ 2.2 and Minitab^®^ 19.1. Spearman’s correlation was used to correlate the expression of cytokines of the humoral and cellular immune responses against seven clinical aspects: Apgar score at 5 min, gestational age, birth weight, invasive mechanical ventilation, noninvasive ventilation, days in the NICU, and days of hospitalization.

## 3. Results

### 3.1. Clinical Aspects

Of the 20 newborns of mothers with COVID-19 and SARS symptoms, 11 were female (55%) and nine were male (45%). Regarding their clinical manifestations, one had enterocolitis (5%), two had asphyxia (10%), two had fever (10%), four had early-onset neonatal sepsis (20%), four had apnea (20%), and seven had respiratory discomfort (35%). By contrast, all 10 newborns in the control group were healthy, as were their mothers.

Biochemical and hematological laboratory tests showed normal levels of alanine aminotransferase, urea and creatinine, troponin, C-reactive protein, saturation, total leukocytes, neutrophils, lymphocytes, and platelets. Meanwhile, aspartate aminotransferase (AST), total and direct bilirubin, creatine phosphokinase and creatine phosphokinase MB fraction, lactate, D-dimer, ferritin, triglycerides, and lactic dehydrogenase (LDH) had abnormally high levels, as detailed in Table 1.

### 3.2. Immunophenotypic Analysis of Cytokines Using Flow Cytometry

We used flow cytometry to identify cytokines, characterize their immunophenotypes, and thereby gather evidence regarding the characteristics of the newborns’ cytokine profiles (Table 2 and Table 3) and cellular profiles.

Identification and quantification using cytometry revealed cytokines of two cellular profiles: Th1 and Th2. Whereas IL-2, TNF, and IFN-γ were identified as representatives of the Th1 profile, IL-6 and IL-10 were identified as representatives of the Th2 profile. The cytokines IL-4 and IL-17A were not identified during the analysis. Once the profile of the expressed cytokines was identified, the averages of the quantified cytokines were calculated for the results shown in Figure 1A and Table 2: IL-6 (143 pg/mL, 30%), TNF-α (103.41 pg/mL, 22%), IFN-γ (95.45 pg/mL, 20%), IL-2 (82.29 pg/mL, 17%), and IL-10 (50.23 pg/mL, 11%).

The control group, however, demonstrated a drastic reduction in the expression of these cytokines, as shown in Figure 1B and Table 3: IL-6 (32%, 31.79 pg/mL), IL-10 (23%, 22.92 pg/mL), IL-2 (23%, 22.87 pg/mL), IFN-γ (14%, 13.21 pg/mL), and TNF-α (8%, 7.92 pg/mL).

Tests were also performed to gauge the levels of cytokine expression in both groups of newborns. TNF demonstrated the highest expression, which was 29.35-fold higher in the neonates of mothers with COVID-19 than in the control group. Among other results, IFN-γ’s expression was 12.94 times higher in the neonates of infected mothers than in the control group, IL-6 was 8.09 times higher, and IL-2 and IL-10 were 6.47 and 4.92 times higher, respectively (Figure 2).

### 3.3. Comparative Analysis of Immune Humoral and Cellular Responses in Newborns of Mothers with COVID-19

Statistical analysis was conducted to identify significant differences in cellular and humoral immune responses between the two groups of newborns. To that end, the SW test was used to gauge the normality of the data and thereby reveal significant results between the groups of newborns and characterize them as abnormal because, in the four conditions tested, only one control group presented data with a normal distribution (Table 4). In what follows, we describe newborns of mothers with COVID-19 as “the infected group” and newborns of healthy mothers as “the control group.”

Comparative analysis using the Mann–Whitney *U* test was also performed between the two groups of newborns. Among the results, the cellular responses between the groups differed significantly. As shown in Table 5, the cytokines corresponding to the cellular response in the control group (median = 18.33 pg/mL) were far lower than those in the infected group (median = 165.56 pg/mL). The table also shows significant differences in the sum of ranks (*R_i_*) as well; in the cellular response of the infected group, the *R_i_* was 390 and had a median of 265.66 pg/mL, whereas it was 45 in the control group and had a median of 18.33 pg/mL (*p* < 0.0001).

When the humoral response was evaluated, significant differences (*p* < 0.0001) in the responses were observed between the control group, with a median of 18.29 pg/mL (*R_i_* = 45), and the infected group, with a median of 169.03 pg/mL (*R_i_* = 390), as shown in Table 6. Thus, there were significant differences in both the cellular and humoral immune responses between the infected and control groups.

Therefore, it can be observed that, during infection, there is an increased production of humoral and cellular immune response cytokines (Figure 3A,B). However, we also observed that the cellular immune response prompted a greater production of cytokines than the humoral response during infection.

Finally, correlations between newborns’ cytokine levels and identified comorbidities were determined. No correlation with statistical significance emerged between Apgar score and humoral response (ρ = 0.05, *p* = 0.8) or cellular response (ρ = 0.2, *p* = 0.4). There was also no significant correlation between cytokines and gestational age and humoral (ρ = 0.05, *p* = 0.8) or cellular response (ρ = 0.1, *p* = 0.7), or between cytokines and birth weight and humoral (ρ = −0.4, *p* = 0.1) or cellular response (ρ = −0.3, *p* = 0.2).

Among other results, days of invasive mechanical ventilation was correlated with humoral response (ρ = 0.07, *p* = 0.8) and cellular response (ρ = −0.1, *p* = 0.6); days of noninvasive ventilation was correlated with cellular (ρ = 0.06, *p* = 0.8) and humoral response (ρ = 0.03, *p* = 0.8); days in the NICU was correlated with humoral (ρ = 0.05, *p* = 0.8) and cellular response (ρ = −0.2, *p* = 0.4); and days of hospitalization was correlated with cellular (ρ = −0.2, *p* = 0.4) and humoral response (ρ = 0.06, *p* = 0.8). Thus, correlations with invasive mechanical ventilation, noninvasive ventilation, NICU, and hospitalization did not show statistical significance.

## 4. Discussion

In humans, COVID-19 infection has a variety of clinical outcomes in forms ranging from asymptomatic to severe [43,44]. Clinical studies have shown that patients with severe COVID-19 experience a significant increase in proinflammatory molecules during the course of the disease—that is, the so-called cytokine storm [27,45,46,47,48]. In response, the immune system not only protects against but also aggravates the disease, and the production of cytokines is essential in directing those responses [25]. In our study, the Th1 and Th2 cytokine profiles of 20 newborns of mothers with COVID-19 were evaluated with the aim of characterizing their cellular and humoral immune responses. All newborns had negative results in the RT-PCR test but were indirectly affected when their mothers were infected, as described above [49].

Due to hormonal changes resulting from pregnancy, the maternal immune system undergoes changes that can prompt negative regulation of the immune system and, in turn, make women vulnerable to infectious disease [50]. To date, conventional SARS-CoV-2 transmission routes have already been found in the pediatric population [51], along with their potential for vertical transmission and the possible effects of the disease during pregnancy [52,53,54,55]. Studies have additionally shown that in newborns, events such as the cytokine storm resulting from maternal inflammatory processes due to COVID-19 can cause serious complications involving damage to the neonate’s organs [56,57]. These aspects were observed during our analysis because all newborns of mothers with COVID-19 did not suffer vertical transmission or present negative RT-PCR test results for SARS-CoV-2. Moreover, when cytokine levels were quantified in the sample, the overexpression of cytokines was verified in the group of newborns of mothers with COVID-19 in relation to the control group.

Our analyses also revealed the expression of cytokines in the Th1 and Th2 profiles. The cytokines produced by Th lymphocytes regulate immunity and inflammatory processes and activate B lymphocytes, the latter of which, representative of humoral immunity, is mediated by antibodies [25,58]. Furthermore, we identified the interleukins IL-2, TNF-α, and IFN-γ, all of which have been described as microbicidal and proinflammatory, as representatives of the Th1 profile. By contrast, cytokines in the Th2 profile, which help with humoral immunity, may be anti-inflammatory and are represented by transforming growth factor β, IL-4, IL-6, IL-10, and IL-13 [59]. However, in our analyses, only IL-6 and IL-10 were identified as representative of the Th2 profile in newborns of mothers with COVID-19.

Among other results, we observed that the cellular immune response of the newborns of mothers with COVID-19 showed an increase in cytokine expression of 803.2% compared with the control group. Of those cytokines, the ones with a Th1 profile were TNF-α, IFN-γ, and IL-2. In past studies, responses with the Th1 profile have been observed in preterm infants affected by human metapneumovirus (HMPV). However, during the quantification of cytokines, it was observed that the levels of IFN-γ/IL-4 were lower in preterm infants affected by HMPV [60]. At the same time, in other studies, researchers have found that babies with HMPV had high levels of CCL5, IFN-γ, and IL-10, all of which represented increased cytokines in the Th1 and Th2 profiles in relation to the cytokines measured in the control group. Even so, premature newborns have minimal amounts of cytokines, such as IFN-γ and IL-4 [61]. It has also been reported that premature babies with IFN-γ deficiency may have more severe symptoms after viral infection, which may lead to comorbidities such as asthma or impaired lung function during their development [62,63,64]. In our study, IFN-γ was 29.32 times more expressed in the neonates of mothers with COVID-19 than in the control group. This molecule is known to relate to macrophage recruitment, inflammation, and protection against infections by intracellular parasites, such as viruses [25].

Research has shown that environments with high concentrations of TNF-α and IFN-γ can induce cell death by way of pyroptosis or apoptosis, either of which increases tissue damage and mortality in cases of COVID-19 [65,66]. In addition, the death of inflammatory cells contributes to the greater release of local cytokines and chemokines, which potentiates the cytokine storm when COVID-19 is present. Studies have shown that patients critically ill with COVID-19 have higher serum concentrations of proinflammatory cytokines and chemokines, including granulocyte colony-stimulating factor, monocyte chemoattractant protein-1, TNF-α, IL-6 and IL-1β, than healthy individuals [24,67].

In other work, babies infected with respiratory syncytial virus showed significantly increased expression of Th1 cytokines, such as TNF-α and IFN-γ and Th2 cytokines, such as IL-6, in relation to the control group [68]. Although IL-6 is part of the inflammatory process, it can be produced by Th2 cells and assist in the activation of B lymphocytes, which are important for the production of antibodies [25]. Similar results were found among infants infected with human bocavirus, including high concentrations of IFN-γ, IL-2, and TNF-α representing the cellular immune response, and IL-4 and IL-10 representing the activation of the humoral immune response [69]. In other research, high levels of Th2-profile cytokines have been associated with respiratory distress from the respiratory syncytial virus [70,71,72], although those respiratory discomforts have also been reported due to an increase in IFN-γ [73]. In our study, levels of IL-6 (143 pg/mL) and IL-10 (50.23 pg/mL) were significantly elevated. Moreover, 35% of newborns of mothers with COVID-19 exhibited respiratory distress, 20% exhibited apnea, and 10% exhibited asphyxia, all possibly due to the high expression of cytokines that affect humoral and cellular immune responses. In previous studies, IL-6 and TNF-α were extremely expressed in newborns who died due to a cytokine storm [74]. The normal values expressed in newborns are 0.0–9.0 pg/mL for IL-6 and 0.0–3.6 pg/mL for TNF-α [75]. However, in newborns of mothers with COVID-19, elevated concentrations of IL-6 (i.e., *M* = 143 pg/mL) and TNF-α (i.e., *M* = 103.41 pg/mL) suggest that the babies were suffering from cytokine storms given dosages of the same molecules in the newborns of healthy mothers. In addition, 20% of neonates of mothers with COVID-19 had sepsis, an event also found in past studies on sepsis in neonates with multisystem inflammatory syndrome [76].

As reported in the literature, an imbalance between IL-10 and IL-6 can lead to excessive inflammation [56]. Beyond that, IL-6 prompts cell differentiation for the TH2 and TH17 profiles and thus participates in systemic inflammation and autoimmune responses [77]. Other studies have also shown that IL-10 balances the amplification of the TH17 response, thereby preventing it from being excessively increased by IL-6 and becoming uncontrollable [78]. For this reason, we propose that IL-10 acts to inhibit the Th17 profile because IL-17A was not detected during cytokine measurement. Also demonstrating this effect, IL-10 possibly regulated the Th17 profile in neonates of mothers with COVID-19 because the high levels of IL-6 detected, as described above, generally act on the excessive increase of the Th17 profile.

The biochemical and hematological laboratory tests of these patients showed normality for ALT, urea, and creatinine, troponin, CRP, saturation, total leukocytes, neutrophils, lymphocytes, and platelets. However, AST, BT, BD, CPK, CPK-MB, lactate, ferritin, D-dimer, LDH, and triglycerides had their parameters increased. Another study showed exactly the same immunological profile as our work, in which they observed increases in AST, LDH, CRP, ferritin, and D-dimers (33). LDH, a common marker of cell death, has also been linked to the pathogenesis of COVID-19. Specifically, high concentrations of LDH in patients’ serum are associated with tissue degradation during COVID-19, and the amount of LDH is considered a powerful predictor of the early recognition of lung injury in cases of COVID-19 [79,80].

Following Can et al.’s finding that neutrophil gelatinase-associated lipocalin is associated with the severity of COVID-19 in pregnant patients [81], the molecule has been regarded as an excellent biomarker for diagnosing the severity of COVID-19 in individuals with the disease. Other studies have shown that, as biomarkers of the severity of COVID-19 in pregnant women, the expression of IL-6 and IL-10 acts by minimizing or maximizing the risks of adverse outcomes, with the lowest expression of the cytokines related to asymptomatic cases and the high expression of severe symptoms [82]. In our study, the elevation of IL-6 and IL-10 was observed in newborns of mothers with COVID-19, who were born inflamed and possibly with a cytokine storm, as mentioned. However, further studies remain necessary to characterize the group of specific cytokines that should be used as biomarkers of inflammation in infants born to mothers with COVID-19.

In our study, we observed the presence of some proinflammatory cytokines in newborns no more than 28 d old, along with increased biochemical tests, even in cases of negative RT-qPCR tests for COVID-19. This hyperinflammatory tendency often prompts the development of viral pneumonia, multisystem inflammatory syndrome in children, and other complications throughout the newborn’s development. Therefore, the post-recovery phase of the clinical condition requires close monitoring of the patient’s development.

Drugs should be given to pregnant women only when the benefit to the mother outweighs the potential risk to the fetus. Fetal drug transfer depends primarily on the permeability of the placental barrier, which forms the interface between fetal and maternal circulation. Basic research on maternal–fetal interaction in SARS-CoV-2 is pivotal to answering many important questions about the prevention and control of COVID-19.

The profile of identified cytokines may be used to better understand the processes that occur in newborns of mothers with COVID-19. Events such as cytokine storms are known to be extremely harmful, even fatal. Therefore, the compression of these events should help direct and advance studies focused on the therapeutic and medication processes in response to the disease.

## 5. Conclusions

In our study, we evaluated the expression of cytokines in newborns of mothers with COVID-19 and compared the levels of those molecules with a group of newborns of healthy mothers.

The application of the CBA kit, together with identification by flow cytometry, was essential to characterize the profile of cytokines expressed by newborn babies of COVID-19 mothers and to compare these data obtained with those of newborns of healthy mothers. Flow cytometry facilitated the characterization of Th1 and Th2 cell profiles. However, IL-17A was not identified, which demonstrates that the inflammation studied was not related to the Th17 profile.

The cytokines TNF-α, IFN-γ, IL-6, IL-2, and IL-10 quantified in newborns of mothers with COVID-19 showed high levels of expression compared with the control group. Furthermore, molecules, such as TNF-α, IL-6, and IL-10, were highly expressed in relation to the control group, thereby indicating the potential for cytokine storms in the newborns of infected mothers.

Regarding cytokines in the Th1 profile, IFN-γ was the most expressive and was probably related to more severe symptoms, including inflammation, asthma, or asphyxia (10%), lung impairment (35%), apnea (20%), and sepsis (20%), in neonates of mothers with COVID-19. The high expression levels of IL-6, TNF-α, and IL-10 may have led to cytokine storms in the neonates of mothers with COVID-19 because the relationship of those cytokines with the event has already been demonstrated in other studies. Clinical manifestations, such as enterocolitis, fever, and alterations in biochemical assays, were present in babies with inflammation, possibly due to cytokine storms.

The cytokines representing the cellular and humoral immune responses showed statistical differences between the two groups of newborns. In addition, during infection, the cellular immune response was outstanding in relation to the expression of cytokines when compared with the humoral immune response. However, factors such as gestational age, birth weight, and Apgar score were not related to the inflammatory processes that we studied. Additional studies are being carried out to understand the role of these cytokines in inflammatory processes in newborn babies. Such an understanding could have an impact on drug and therapeutic improvements in the pediatric population.

## Figures and Tables

**Figure 1 biomedicines-11-00910-f001:**
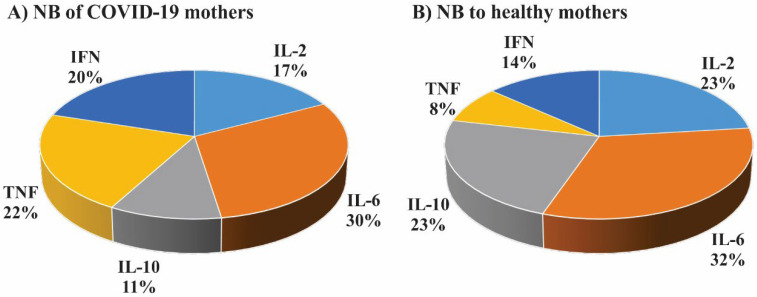
Analysis of cytokine percentages after quantification using flow cytometry. (**A**) Percentage of cytokines expressed by newborns (NB) of mothers with COVID-19. (**B**) Percentage of cytokines expressed by newborns of healthy mothers.

**Figure 2 biomedicines-11-00910-f002:**
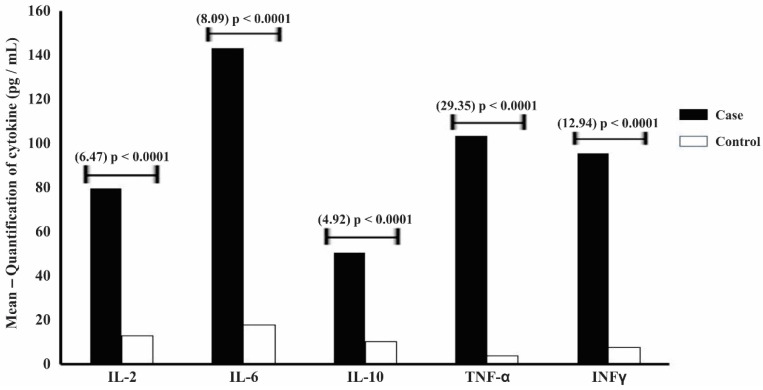
Comparative analysis of cytokine expression in newborns of mothers with COVID-19 and newborns of healthy mothers. Numbers in columns represent cytokine quantification in picograms per milliliter (pg/mL). NB (newborn).

**Figure 3 biomedicines-11-00910-f003:**
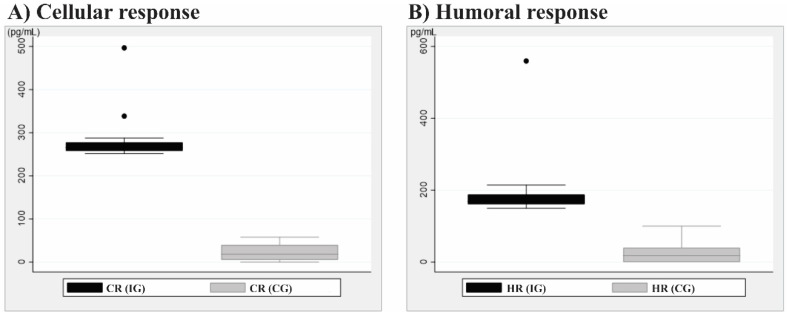
Analysis of immune responses identified in newborns. (**A**) Evaluation of cellular response (CR) cytokine levels in relation to newborns of mothers with COVID-19 (IG) and newborns of healthy mothers (CG). (**B**) Assessment of humoral response (HR) cytokine levels in relation to newborns of mothers with COVID-19 (IG) and newborns of healthy mothers.

**Table 1 biomedicines-11-00910-t001:** Chief clinical manifestations and laboratory test results in newborns (*n* = 20) of mothers with COVID-19.

Sex	Total (%)
Male	45%
Female	55%
**Clinical manifestations**	**Total (%)**
Enterocolitis	5%
Early-onset neonatal sepsis	20%
Asphyxia	10%
Cough	5%
Fever	10%
Apnea	20%
Respiratory discomfort	35%
Need for NICU	100%
Deaths	0%
**Laboratory tests**	**Average (min. and max.)**
Aspartate aminotransferase	56 (26–78)
Alanine aminotransferase	11.77 (0.8–21)
Total bilirubin	6.04 (2.8–11.8)
Direct bilirubin	1.16 (0.3–5.7)
Urea	24.64 (7–42)
Creatinine	0.76 (0.5–1.3)
Creatinophosphokinase	445.64 (109–1.197)
CK-MB	44.81 (23–55)
Troponin	All negative
Lactate	33.77 (4.1–72.6)
D-dimer	585 (1.06–1066)
PCR	0.22 (0.1–0.7)
Lactate dehydrogenase	596.9 (457–1036)
Ferritin	245.65 (150.32–558.71)
Triglycerides	118.72 (43–239)
RT-qPCR	All negative
Oxygen saturation	98.97 (98.2–99.7)
Total leukocytes	10.790 (5.500–18,600)
Neutrophils	5.433 (1.975–12,640)
Lymphocytes	3.975 (1.637–9.343)
Platelets	205.384 (14.500–372.000)

Note. CK-MB = creatine kinase subunits M and B; NICU = neonatal intensive care unit.

**Table 2 biomedicines-11-00910-t002:** Quantification of cytokines in newborns of mothers with COVID-19.

Quantification of Cytokines (pg/mL)					
Patient ID	IL-2	IL-6	IL-10	TNF-α	IFN-γ
1	74.51	136.26	51.21	103.25	100.2
2	79.69	141.95	46.63	101.22	96.36
3	77.94	127.59	38.85	95.38	92.14
4	73.15	126.83	38.73	94.71	91.16
5	71.43	128.19	43.43	99.06	91.37
6	79.98	113.69	47.97	97.73	91.13
7	68.17	114.84	35.14	92.36	91.15
8	74.00	131.53	44.31	99.74	92.11
9	74.17	119.49	59.92	100.97	87.29
10	181.14	405.94	153.85	188.58	127.17
11	105.13	163.6	50.98	116.16	117.28
12	70.54	116.88	43.47	96.23	86.70
13	69.51	115.16	39.85	94.23	89.18
14	74.22	119.4	41.34	99.75	95.17
15	72.78	122.55	43.23	94.69	89.43
16	90.59	142.12	44.04	101.38	95.72
17	72.01	119.04	39.25	93.33	88.66
18	72.76	121.61	41.28	96.34	95.51
19	82.15	152.78	51.35	103.13	97.33
20	81.85	140.45	49.78	99.96	93.84

Note. IL-2 = interleukin-2; IL-6 = interleukin-6; IL-10 = interleukin-10; TNF-α = tumor necrosis factor alpha; IFN-γ = interferon gamma; pg/mL = picograms per milliliter.

**Table 3 biomedicines-11-00910-t003:** Quantification of cytokines in the newborns of healthy mothers.

Quantification of Cytokines (pg/mL)					
Patient ID	IL-2	IL-6	IL-10	TNF-α	IFN-γ
1	15.86	30.41	20.02	ND	20.01
2	ND	60.08	40.18	6.17	12.16
3	20.08	ND	ND	20.13	ND
4	ND	40.15	ND	ND	ND
5	ND	ND	18.29	ND	3.73
6	30.26	20.18	13.21	2.26	25.42
7	40.05	8.17	ND	ND	ND
9	ND	ND	ND	ND	5.05
10	8.13	ND	ND	3.15	ND

Note. IL-2 = interleukin-2; IL-6 = interleukin-6; IL-10 = interleukin-10; TNF-α = tumor necrosis factor alpha; IFN-γ = interferon gamma; pg/mL = picograms per milliliter; ND = not detected.

**Table 4 biomedicines-11-00910-t004:** Significant results of the Shapiro–Wilk (SW) test for qualifying the distribution as abnormal between the humoral response (HR) and cellular response (CR) presented by the control group (CG) and the infected group (IG).

SW Normality Test	CR (IG)	HR (IG)	CR (CG)	HR (CG)
*n*	20	20	9	9
*M*	281,143	193,226	23,606	27,854
*SD*	54,272	87,916	20,468	33,054
*W*	0.494	0.402	0.908	0.840
*p*	0.0057	0.0046	0.3626	0.0693

**Table 5 biomedicines-11-00910-t005:** Comparison of cellular immune response between the control and infected groups using the Mann–Whitney *U* test.

Mann–Whitney *U* Test	Cellular Response (Infected Group)	Cellular Response (Control Group)
Sample size	20	9
Sum of ranks (*R_i_*)	390	45
Median	265.66	18.33
*U*	0	-
*z* (*U*)	42.426	-
*p* (one-tailed)	<0.0001	-
*p* (two-tailed)	<0.0001	-

**Table 6 biomedicines-11-00910-t006:** Comparison of humoral immune response between the control and infected groups using the Mann–Whitney *U* test.

Mann–Whitney *U* Test	Humoral Response (Infected Group)	Humoral Response (Control Group)
Sample size	20	9
Sum of ranks (*R_i_*)	390	45
Median	169.03	18.29
*U*	0	-
*z* (*U*)	42.426	-
*p* (one-tailed)	<0.0001	-
*p* (two-tailed)	<0.0001	-

## Data Availability

The authors confirm that all data supporting the findings of the study are available in the article.

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
