# Peer review of "Characterization of T Helper 1 and 2 Cytokine Profiles in Newborns of Mothers with COVID-19"

_biomedicines, 2023, doi:10.3390/biomedicines11030910_

Round 1

Reviewer 1 Report

The article has some powerful parts such as the introduction, the sample and methods section and the statistical analysis. However, it is difficult to draw clear conclusions from the discussion and the conclusions section about the clinical implication of the study. There should be a paragraph for the clinical significance of the findings and how these may be useful in practice. Should we estimate these inflammatory mediators in the neonates in order to take action and improve their outcome? Is this adding extra information over clinical and standard laboratory evaluation of the affected neonates. I also think that it would be more interesting to correlate the levels of these mediators to the clinical picture of the neonates, instead of comparing them with normal cases (this correlation may be possible with the data already gathered). Finally, there should be a paragraph dedicated to how we can differentiate the origin of these inflammatory mediators (neonatal? maternal?). Is the comparative level of increase helpful? Could there be placental origin for some of these mediators (see literature on placentas of affected mothers)?

Optional: There could be a graph showing the inflammatory response and pathway

Author Response

#Reviwer 1

The article has some powerful parts such as the introduction, the sample and methods section and the statistical analysis. However, it is difficult to draw clear conclusions from the discussion and the conclusions section about the clinical implication of the study.

Note: We would like to thank you for your comments and suggestions.

Question 1: There should be a paragraph for the clinical significance of the findings and how these may be useful in practice.

Answer: This samples collected from newborns of mothers with COVID-19 and SARS symptoms were the first variants of SARS-CoV-2. These were collected in the first wave of COVID-19 that occurred in Brazil, between April and May 2020. During this collection period, there were no mothers immunized through vaccination programs, since there were no vaccines for COVID-19 . In addition, another factor that contributed to the low number of samples was the Lockdown decree in the country that started on April 20, 2020. This event resulted in a large decrease in the flow of people and access to hospitals, since many of the patients infected with SARS-CoV-2 chose to remain in seclusion. Others samples were not collected in the post-lockdown period due to most mothers having already been immunized with COVID-19 vaccines. Therefore, we obtained N from 20 sick patients and decided to proceed with the analyses.

Question 2: Should we estimate these inflammatory mediators in the neonates in order to take action and improve their outcome? Is this adding extra information over clinical and standard laboratory evaluation of the affected neonates.

Answer: We cannot confirm this in the current study. However, we are continuing the investigation into the expression of cytokines in the pediatric population affected by SARS/COVID-19.

Question 3: I also think that it would be more interesting to correlate the levels of these mediators to the clinical picture of the neonates, instead of comparing them with normal cases (this correlation may be possible with the data already gathered).

Answer:  Yes, as suggested, we tried to correlate the levels of mediators with the clinical picture of the neonates. However, we did not obtain significant results for this correlation, as shown below.

Table 1 presents the description of the data referring to the group of neonates with positive mothers for COVID-19. Newborns of COVID-19 mothers were divided into severe cases and mild cases. Subsequently, comparisons were made between the groups obtained from affected individuals.

Table 1. Descriptive statistics of infected group data.

Serious cases (n = 6)

Mild cases (n = 11)

Variable

Mean

SE

SD

Median

Mean

SE

SD

Median

IL-2

89.7

18.3

44.8

72.1

78.35

2.92

9.68

74.51

IL-6

168

47.6

116.7

121

131.15

4.87

16.7

127.59

IL-10

63.2

18.4

45.2

43.4

44.82

1.51

5.01

44.31

TNF

112

15.4

37.6

97.6

100.02

1.91

6.33

99.74

IFN

95.52

6.38

15.6

90.29

95.92

2.39

7.92

95.17

Cellular response

297.2

40

97.9

259.4

274.29

7

23.2

268.84

Humoral Response

231.2

65.9

161

168.7

175.97

6.05

20.1

166.44

APGAR 5

5.167

0.87

2.14

4.5

9.273

0.30

1.01

10

GA

237.3

10.4

25.6

232

253.36

8.32

27.6

269

BW

2101

279

684

2043

2323

220

729

2290

IMV - DAYS

10.33

4.96

12.1

5.5

4.27

2.87

9.53

0

NIV-DAYS

3.5

1.15

2.81

3

4.09

2.22

7.35

0

ICU-DAYS

21.5

4.94

12.1

19.5

11

5.42

17.98

1

HAR-DAYS

34

4.84

11.87

35

20.36

9.1

30.17

4

All possible comparisons between the variables in Table 1 were performed. However, only two variables showed a significant difference between the medians, with the median APGAR being lower in mild cases, showing a reduced difference with -5 [-6 to -2; p-value = 0.003] and number of days with invasive mechanical ventilation (IMV) demonstrating higher in severe cases with 4 [2 to 12; p-value = 0.01]. Comparisons between variables can be observed through the analysis of Figures 1 and Figure 2.

Figure 1. Presentation of variables in a box plot: group of severe cases.

Figure 2. Presentation of variables in a box plot: group of mild cases.

Considering the newborns of mothers infected with COVID-19 (regardless of severity), correlations were performed between the variables: APGAR, GA, BW, IMV-DAYS and NIV-DAYS. APGAR showed a positive and moderate correlation with GA, that is, as gestational age increased, the APGAR score also increased (r = 0.52; p-value = 0.03). APGAR showed a moderate negative correlation with NIV-DAYS and IMV-DAYS, respectively (r = - 0.65; p-value = 0.004 and r = - 0.67; p-value = 0.003), thus, the increase in days in both ventilation procedures, reduces the APGAR score. There was no correlation between APGAR and GW. There was a strong negative correlation between NIV-DAYS and the GA and BW variables, respectively (r = - 0.9; p-value < 0.001 and r = 0.8; p-value < 0.001). Thus, reduced gestational age and reduced newborn weight increase the number of days with VIN. There was a moderate positive correlation between NIV-DAYS and IMV-DAYS (r = 0.68; p-value = 0.002). The IMV-DAYS showed a moderate negative correlation with GA and BW, respectively (r = - 0.53; p-value = 0.02 and r = -0.55; p-value = 0.02). Thus, the reduced gestational age and the reduced weight of the newborn increase the number of days with IMV. Gestational age and newborn weight were positively correlated (r = 0.8; p-value < 0.001). The correlations can be seen in Figure 3.

Figure 3. Correlation between clinical variables of neonates of mothers with COVID-19.

Question 4: Finally, there should be a paragraph dedicated to how we can differentiate the origin of these inflammatory mediators (neonatal? maternal?). Is the comparative level of increase helpful? Could there be placental origin for some of these mediators (see literature on placentas of affected mothers)?

Answer: We conducted a general search to answer your question. However, most of the data described in the literature end up describing inflammatory mediators expressed by certain diseases or conditions. Therefore, there are no studies that allow us to conclude or support this doubt.

Optional: There could be a graph showing the inflammatory response and pathway

Answer: We were unable to assemble a graph of the inflationary response corresponding to the study. In this study, we only carried out the identification of cytokines in the serum of the patients under study. However, we set up a graphic abstract to better illustrate the work.

Reviewer 2 Report

Thank you very much for giving me the opportunity to review this study. My specific comments regarding the manuscript are included below;

- There are some grammatical errors that should need to be corrected (i.e. being one of the main groups that present a risk of, and may have symptoms, infection of other viruses, cytokine levels with a control group born to healthy mothers,  missing commas, etc.).

- All abbreviations should have been provided in full on the first mention and this applies to the title, running (short) title, abstract, impact statement, main text, and each table/figure independently as they will be read independently. Please use the abbreviations correctly and effectively. So, all the abbreviations should be checked (i.e. SARS, etc.).

- ‘In addition, the pediatric public is the most affected by SARS worldwide, generating about 1.9 million deaths per year, with 70% of these deaths occurring in developing countries and 30% of child deaths resulting from this infection.’’As already reported in other studies, it is known that the virus affects the pediatric population, being this one of the main groups that present risk of causes of death, including newborns.’ These sentences are similar. In scientific documents, sentences should be to the point and authors should avoid repetition.

- Please define the designation of the study (retrospective, prospective, etc.).

- ‘enterocolitis was diagnosed according to the Bell classification. Neonatal early sepsis was considered in the first 72 h of life, with maternal infectious risk and clinical and laboratory changes of newborn, following the standard hospital service protocol. To evaluate asphyxia, the APGAR score of 5' ≤ 5 was considered, or when the  newborn required positive pressure ventilation associated with changes in arterial blood gas analysis. Apnea was considered when it could cause respiratory pause ≥ 20 seconds, followed by bradycardia and drop in saturation. Respiratory distress was diagnosed and classified according to Silverman-Andersen Retraction Score (SAs).’ Please cite these sentences.

- Please define birth asphyxia more specifically (Tunç Åž, OÄŸlak SC, Gedik Özköse Z, Ölmez F. The evaluation of the antepartum and intrapartum risk factors in predicting the risk of birth asphyxia. J Obstet Gynaecol Res. 2022;48(6):1370-1378). 

- Inclusion and exclusion criteria are not clear. It should be specified. Please write the features of the study groups more comprehensively.

- A recent study concluded that NGAL was highly associated with COVID-19 severity. NGAL might be a useful biomarker to diagnose the disease severity in patients with COVID-19 (Can E, OÄŸlak SC, Ölmez F, Bulut H. Serum neutrophil gelatinase-associated lipocalin concentrations are significantly associated with the severity of COVID-19 in pregnant patients. Saudi Med J. 2022;43(6):559-566). Please mention it in your main text and cite this article.

- How did you decide to enroll 30 cases in the study cohort? Did you perform a sample size calculation?

- Please write the strengths and limitations of this study.

- Practical implications and future research direction are not mentioned. Please discuss the generalisability (external validity) of the study results. 

Author Response

#Reviwer 2 - Thank you very much for giving me the opportunity to review this study. My specific comments regarding the manuscript are included below;

Note: We would like to thank you for your comments and suggestions.

Question 1: There are some grammatical errors that should need to be corrected (i.e. being one of the main groups that present a risk of, and may have symptoms, infection of other viruses, cytokine levels with a control group born to healthy mothers,  missing commas, etc.).

Answer: Thanks for the note. The entire article has been proofread and corrected.

Question 2: All abbreviations should have been provided in full on the first mention and this applies to the title, running (short) title, abstract, impact statement, main text, and each table/figure independently as they will be read independently. Please use the abbreviations correctly and effectively. So, all the abbreviations should be checked (i.e. SARS, etc.).

Answer: The requested changes were made. Thus the acronyms were included in the first citation of each term during the writing of the file: COVID-19, (Corinavirus Disease 2019), GM-CSF (Granulocyte Macrophage Colony-Stimulating Factor), ICU (Neonatal Intensive care unit), IFN-γ (Intereferon-gamma), IL (Interleukin), NB (Newborn), RT-PCR (Real-time Polymerase Chain Reaction), SARS (Severe Acute Respiratory Syndrome), SARS-CoV-2 (Severe Acute Respiratory Syndrome Coronavirus 2), SW test (Shapiro-Wilk test), Th (T helper), TNF-a (Tumor Necrosis Factor-alpha).

Note: Our team thought of not using the acronyms in the title and abstract, only in the body of the text (introduction, methods, results and discussion), but we will accept your suggestion.

Question 3: ‘In addition, the pediatric public is the most affected by SARS worldwide, generating about 1.9 million deaths per year, with 70% of these deaths occurring in developing countries and 30% of child deaths resulting from this infection.’’As already reported in other studies, it is known that the virus affects the pediatric population, being this one of the main groups that present risk of causes of death, including newborns.’ These sentences are similar. In scientific documents, sentences should be to the point and authors should avoid repetition.

Answer: As requested, the sentence has been restructured to be direct and avoid similar sentences and become more understandable and fluid.

Question 4: Please define the designation of the study (retrospective, prospective, etc.).

Answer: It was a prospective cohort study, since cytokine samples were obtained from newborns of mothers positive for COVID-19. This was included at the beginning of topic item 2.3 (Collection and processing of samples).

Question 5: ‘enterocolitis was diagnosed according to the Bell classification. Neonatal early sepsis was considered in the first 72 h of life, with maternal infectious risk and clinical and laboratory changes of newborn, following the standard hospital service protocol. To evaluate asphyxia, the APGAR score of 5' ≤ 5 was considered, or when the  newborn required positive pressure ventilation associated with changes in arterial blood gas analysis. Apnea was considered when it could cause respiratory pause ≥ 20 seconds, followed by bradycardia and drop in saturation. Respiratory distress was diagnosed and classified according to Silverman-Andersen Retraction Score (SAs).’ Please cite these sentences.

Answer: As suggested, references were cited that demonstrate the methodology used for each of the analyzes used in the study. The citation is shown below.

- Bell, M.J.; Ternberg, J.L.; Feigin, R.D.; Keating, J.P.; Marshall, R.; Barton, L.; Brotherton, T. Neonatal necrotizing enterocolitis. Therapeutic decisions based upon clinical staging. Annals of surgery 1978, 187, 1.

- Sankar, M.J.; Agarwal, R.; Deorari, A.K.; Paul, V.K. Sepsis in the newborn. The Indian Journal of Pediatrics 2008, 75, 261-266.

- Manganaro, R.; Mamì, C.; Gemelli, M. The validity of the Apgar scores in the assessment of asphyxia at birth. European Journal of Obstetrics & Gynecology and Reproductive Biology 1994, 54, 99-102.

- Gilstrap III, L.C.; Leveno, K.J.; Burris, J.; Williams, M.L.; Little, B.B. Diagnosis of birth asphyxia on the basis of fetal pH, Apgar score, and newborn cerebral dysfunction. American journal of obstetrics and gynecology 1989, 161, 825-830.

- Aggarwal, R.; Singhal, A.; Deorari, A.K.; Paul, V.K. Apnea in the newborn. The Indian Journal of Pediatrics 2001, 68, 959-962.

- Silverman, W.A.; Andersen, D.H. A controlled clinical trial of effects of water mist on obstructive respiratory signs, death rate and necropsy findings among premature infants. Pediatrics 1956, 17, 1-10.

Question 6: Please define birth asphyxia more specifically (Tunç Åž, OÄŸlak SC, Gedik Özköse Z, Ölmez F. The evaluation of the antepartum and intrapartum risk factors in predicting the risk of birth asphyxia. J Obstet Gynaecol Res. 2022;48(6):1370-1378).

Answer: In our methodology we include how asphyxia was characterized. We considered asphyxia in newborns (of mothers with COVID-19) with APGAR ≤5, need for positive pressure ventilation and need for intubation. These criteria were used for newborns of COVID-19 mothers, since healthy patients did not present these symptoms mentioned above.

Question 7: Inclusion and exclusion criteria are not clear. It should be specified. Please write the features of the study groups more comprehensively.

Answer: The Inclusion criteria were changed as suggested. It has been worded more comprehensively as requested.

Question 8: A recent study concluded that NGAL was highly associated with COVID-19 severity. NGAL might be a useful biomarker to diagnose the disease severity in patients with COVID-19 (Can E, OÄŸlak SC, Ölmez F, Bulut H. Serum neutrophil gelatinase-associated lipocalin concentrations are significantly associated with the severity of COVID-19 in pregnant patients. Saudi Med J. 2022;43(6):559-566). Please mention it in your main text and cite this article.

Answer: We observed that among the patients studied, the babies of mothers with COVID-19 were born with symptoms of inflammation when compared to babies in the control group. However, our study has N as a limitation, which was small due to the sample collection period (first variant of the virus and the samples were obtained in the first semester of the pandemic). However, it was not possible to compare the final outcome of severity or whether there was expression of a specific group of cytokines. With that, we are continuing the investigation.

Note: We have included the requested citation in the discussion.

Question 9: How did you decide to enroll 30 cases in the study cohort? Did you perform a sample size calculation?

Answer: The type of samples (newborns from mothers who were infected with COVID-19 and who showed symptoms of SARS) were collected in the first wave of COVID-19 that occurred in Brazil (Between April and May 2020), being considered the first variant of SARS-CoV-2. It is noteworthy that there were no mothers immunized by vaccination in this period of collection of samples from infected patients. Therefore, this made it impossible to obtain samples on a large scale. Another factor that contributed to the low N of samples was the Lockdown decree in the country (April 20, 2020), which resulted in a large decrease in the flow of people and access to hospitals, since many infected individuals chose to stay in seclusion. In addition, the periods after the lockdown, the mothers had already been immunized, which could interfere with the future results that would be analyzed. That's why we reached the number of 20 sick patients and decided to proceed with the analyses.

Question 10: Please write the strengths and limitations of this study.

Answer: As a strong point of the study, it should be noted that the cytokine samples were collected during the first wave of COVID-19 in the country, where, in a short time, a state of Lockdown was declared and little was known about the disease, ways of therapy (drugs, corticoids, etc...) and diagnosis.

As a weak point, due to the low number of samples, it was not possible to carry out more robust analyzes and verify whether the expression of these cytokines had an influence on the severity of the babies. Although they were born with high expression of cytokines, we were unable to interpret the outcome because of N. However, we have more studies in progress to answer these questions.

Question 11: Practical implications and future research direction are not mentioned. Please discuss the generalisability (external validity) of the study results.

Answer: As implications and future research directions, it is emphasized that the profile of cytokines identified may be used for a better understanding of the inflammatory processes that occur in newborn babies of COVID-19 mothers. Events such as cytokine storms are known to be extremely harmful and even fatal. Therefore, the compression of these events will be able to direct and help in the advancement of studies focused on the therapeutic and medication processes of this disease. In addition, additional studies are being carried out in order to answer the current questions of this study. This information has been inserted at the end of the discussion.

Round 2

Reviewer 2 Report

I consider this manuscript can be published and no need for further revision.